# Imaging in-operando LiCoO$_2$ nanocrystallites with Bragg coherent X-ray diffraction
David Serban [1,2] ✉, Daniel G. Porter[2], Ahmed H. Mokhtar [1], Mansoor Nellikkal[1], Sivaperumal Uthayakumar[3,4], Min Zhang[5], Stephen P. Collins[2], Alessandro Bombardi[2], Peng Li[2], Christoph Rau[2] & Marcus C. Newton[1] ✉

Although the LiCoO$_2$ (LCO) cathode material has been widely used in commercial lithium ion batteries (LIB) and shows high stability, LIB's improvements have several challenges that still need to be overcome. In this paper, we have studied the *in-operando* structural properties of LCO within battery cells using Bragg Coherent X-ray Diffraction Imaging to identify ways to optimise the LCO batteries' cycling. We have successfully reconstructed the X-ray scattering phase variation (a fingerprint of atomic displacement) within a $\approx (1.6 \times 1.4 \times 1.3) \mu m^3$ LCO nanocrystal across a charge/discharge cycle. Reconstructions indicate strained domains forming, expanding, and fragmenting near the surface of the nanocrystal during charging, with a determined maximum relative lattice displacements of 0.467 Å. While discharging, all domains replicate in reverse the effects observed from the charging states, but with a lower maximum relative lattice displacements of 0.226 Å. These findings show the inefficiency-increasing domain dynamics within LCO lattices during cycling.

Li-ion batteries (LIB) are unmatchable in terms of energy and power densities, which have made them good candidates for portable electronics, electric power tools and both hybrid and full electric vehicles[1–3]. LIBs can help to reduce the world-wide dependence on fossil fuels[4–6]. Though rechargeable LIBs have had many proposed technologies[7–11], a particular one that involves the use of graphite as an anode material[12,13] and a mixture of lithium and transition metal-based oxides has become the prevalent technology[14]. LiCoO$_2$ (LCO) introduced by Goodenough[15] is both the first and the most commercially successful layered transition metal oxide cathode. The practical and theoretical capacities of the cathode materials (170–275 Ah/kg)[4,16] are generally lower than the theoretical capacity of the anode material (372 Ah/kg)[17–21]. Therefore, LIB-improving research is focused on increasing the capacity by replacing LCO with other compounds[4,22,23]. Despite LCO being widely used in commercial LIBs and showing high stability[15], its practical capacity is limited ( ~155 Ah/kg)[4,16]. Furthermore, the low thermal stability[24–27] (which can result in combustion[28–30]), high cost[31–33] and fast capacity fade at high current rates or during deep cycling are major limitations of LCO.

Bragg Coherent X-ray Diffraction Imaging (Bragg CDI) is a lensless far field imaging technique that allows imaging of nanometre scale crystalline materials with a sensitivity to sub-Angstrom displacements[34]. It is largely non-destructive and can provide strain information at the surface and throughout the bulk of a material. Conventional Bragg CDI is performed by illuminating a sample with a spatially coherent X-ray source where the coherence length exceeds the dimensions of the crystal[34–36]. In Bragg reflection geometry, scattered light from the entire volume of the crystal interferes in the far-field, producing a three-dimensional diffraction pattern[37]. Iterative phase reconstruction methods can then recover the complex three-dimensional electron density and phase information[38–40]. The displacement of ions throughout the bulk is directly related to the phase and can be used to obtain strain information according to the relation $\phi = \mathbf{Q} \cdot \mathbf{u}$ (where $\mathbf{u}$ and $\mathbf{Q}$ are the atomic displacement and momentum transfer, respectively)[34,41,42]. The Bragg CDI method is ideally suitable for the structural changes study within battery materials such as LCO, as it is able to provide three dimensional atomic displacement and strain information with Angstrom sensitivity. Early work on Bragg coherent diffraction imaging of cathode battery materials has demonstrated the techniques' feasibility providing impetus for further investigation[22,23,43,44].

Shabalin et al.[44] performed a Bragg CDI experiment previously on LiNi$_{0.5}$Mn$_{1.5}$O$_4$, but have encountered difficulties with particle overlap due to the relatively large beam size and the chosen lower-order reflection. Ulvestad et al.[43] have successfully reconstructed nanocrystals of

[1]Department of Physics & Astronomy, University of Southampton, Southampton, UK. [2]Diamond Light Source, Harwell Oxford Campus, Didcot, Oxfordshire, UK. [3]Department of Physics, Royal Holloway, University of London, Egham, UK. [4]ISIS Pulsed Neutron and Muon Source, STFC Rutherford Appleton Laboratory, Didcot, Oxfordshire, UK. [5]School of Chemistry, University of Southampton, Southampton, UK. ✉e-mail: das1g13@soton.ac.uk; m.c.newton@soton.ac.uk

$LiNi_{0.5}Mn_{1.5}O_4$ and showed interesting strain inhomogeneity formation. Estandarte et al.[22] have successfully reconstructed $LiNi_{0.8}Mn_{0.1}Co_{0.1}O_2$ (NMC811) nanocrystal from the pristine stage, undergoing charging and slight discharging. A comprehensive study performed by Liu et al.[23] has attempted to image the strain within Li- and Mn-rich cathode materials at very fine increments in charging voltages, however, likely due to the instability associated with Mn-based cathode materials, they were unable to obtain diffraction data during discharging stages. Despite the previous studies covering a range of Li-ion based cathode materials, none were found focusing on LCO, which is most commonly found in electronic devices.

In the following, we successfully determine how crystalline domains are forming and evolving within a single grain of LCO inside a windowed electrochemical cell at a range of voltages. The diffraction patterns were obtained by performing Bragg CDI on a coin cell featuring a window for X-rays at different voltages, with reconstructions employing a machine learning algorithm. The results present features that are state dependent and contain reproducible strains found mostly on the nanocrystal's surface, therefore suggesting possible Li migration pathway formation and variation, as well as Li-depletion stability when fully charged.

## Results

The experiment was performed on a novel LCO coin cell design as described in the 'Methods' section. The Bragg CDI measurements were performed on the I13-1 beamline at Diamond Light Source synchrotron facility using 13.5 keV X-rays. The experimental setup is illustrated in Fig. 1 and described in detail in the 'Methods' section. As seen in Fig. 2, the reconstructions resulting from the machine learning algorithm (see 'Methods' section) are morphologically similar to one another. Table 1 contains the calculated $\chi^2$ values for each of the reconstructions. S2 in the Supplementary contains explanations and discussions of the measured currents during all the cycling stages. Furthermore, S3 in the Supplementary describes the slight shifts in the scattering angles. Regarding the reconstructions, the real-space constraints applied to each of the reconstruction attempts were identical, and the amplitudes are similar across reconstructions. The principal component analysis (PCA) performed on the reconstructions to quantify the phase changes was performed as detailed in the 'Methods' section.

The nanocrystal was determined to be of $\approx (1.6 \times 1.4 \times 1.3)$ µm³ in size ($\pm 0.1$ µm in each direction) with a resolution of 49.5 nm. The reconstruction of the crystal shown in Fig. 2I corresponds to the nanocrystal after charging

with 2.5 V and displays missing densities on the surface and core. Other reconstructions fill the missing densities completely (Fig. 2II, III), partially (Fig. 2IV), or partially but lacking in other regions (Fig. 2.IV). Considering all reconstructions are of the same crystal at different stages, the missing densities from each individual reconstruction needed to be filled based on the other reconstructions.

As shown in Fig. 3a, the same reconstructions have had the missing densities filled (S4 in Supplementary). The complexity of the observed phase patterns, however, requires extended descriptions and discussions of the phenomena to fully understand the domain dynamics.

Figure 3aI's slice at 2.5 V shows that the crystal's phase shifts continuously across the crystal. Domains form on all the surfaces except for the bottom left corner. The peripheral lower left of the nanocrystal shares most of the phase with a significant portion of the core.

From the same perspective, Fig. 3aIV shows the reconstruction slice when charging with 4 V that shares some features with Fig. 3aI: the uniformity of the phase found in the core and on the surface is more pronounced in Fig. 3aIV; a region within the nanocrystal presents an abrupt phase wrap that is almost parallel to the one found in Fig. 3aI. Unlike Fig. 3aI, the upper left domain has degenerated into multiple smaller regions of phase, and the lower left region shows a higher conformity with the phase found within the core region.

Figure 3aII shows the slice through the reconstruction during the 3 V charge. Phase shifts are much stronger and complex than in Fig. 3aI. The central domain is largely uniform while the edge domains display increased activity: the right-most domain migrated and tilted around the X-axis (right hand rotation) by $\approx 35°$; the upper-left domain has expanded along the upper surface; a bottom-left domain has formed.

Similarly, Fig. 3aIII shows a slice through the reconstruction after charging with 3.5 V. Though not as strong as in Fig. 3aII, phase shifts are apparent: top-left domain expanded towards the central and the bottom-left domains of the crystal; the bottom-left domain has expanded towards the central domain; right-most domain split into two domains.

Figure 3aV contains a slice and the phase within the nanocrystal after beginning to discharge with 3.5 V—please note the alignment of the subfigures to facilitate the visual comparison of the same voltage applications at different stages. Figure 3aV shows a similar degree of phase shifts as Fig. 3aIII following the Fig. 3aIV fully charged reconstruction, but with slight changes: the upper left-domain is still disintegrated; the bottom-left domain shows some uniformity mixed with strain as well; the right-most domain has tilted around the X-axis by $\approx 35°$ counter-clockwise.

Finally, Fig. 3aVI shows the phase within after discharging with 3.5 V. The upper-left domain reformed similarly to its structure in Fig. 3aII with some visible phase shifts across shorter distances. The lower-left domain reduced in size, but larger than in Fig. 3aII. The right-most domain is mostly unchanged.

The collage shown in Fig. 3a also contains the rectilinear axes system of the real-space laboratory frame of reference in which the scattering vector (Q) is of the magnitude $\approx 3.398 \cdot 10^{10}$ m$^{-1}$ approximately along the X-axis of the rectilinear system (S6.i in the Supplementary). In other words, the point of view chosen for the visualisation of the phase should be approximately on the scattering vector, such that it comes mostly out of the page, therefore, the X-axis is along the scattering vector and the Z-axis is along the incident X-ray beam. The slice through the objects are also all across and perpendicular to the X-axis. The observed phases, therefore, are the components of the atomic displacements along the into-the-page to out-of-the-page direction.

As the crystal lattice orientation cannot be determined from a single reflection, any strain could only be calculated along the singular scattering vector. Phase changes between $-\pi$ and $\pi$ can only be obtained with displacements' components along the Q-vector (S6.i in the Supplementary). The mean phase variations between any state's reconstruction and the initial reconstruction of the 2.5 V state can further be correlated to a mean relative atomic displacement and implicitly a mean relative strain (Table 1). These calculations are done by altering Eq. (5) to determine the strain covariant

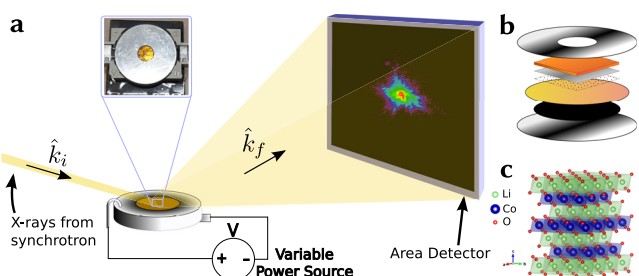

**Fig. 1 | Experimental Setup and Sample Characteristics. a** Schematic of the experimental setup, where incoming coherent X-rays from the synchrotron source pass through the window of the coin cell, are scattered by the nanocrystals in the cathode, and pass through the window towards the detector in the Fraunhofer condition. The coin cell is connected to a circuit maintaining a constant voltage. **b** Schematic of the coin cell sample including (from top to bottom) the drilled-through cathode steel cap, Kapton window for rigidity, evaporated and condensed 2.2 µm Al layer, monodispersed LCO nanocrystals, Celgard 2400 monolayer microporous membrane as a separator, graphite anode, anode steel cap. The cathodic space is occupied by the assembly of window-substrate with LCO nanocrystals, ensuring the connection between the steel cap and the cathode material with strips of Al tape connected to conductive faces. Everything in the interior is submerged in electrolyte. **c** LCO crystal structure of multiple unit cells distributed within the a-b plane to highlight the layers, characteristic of a 2D battery material illustrated using Vesta[54].

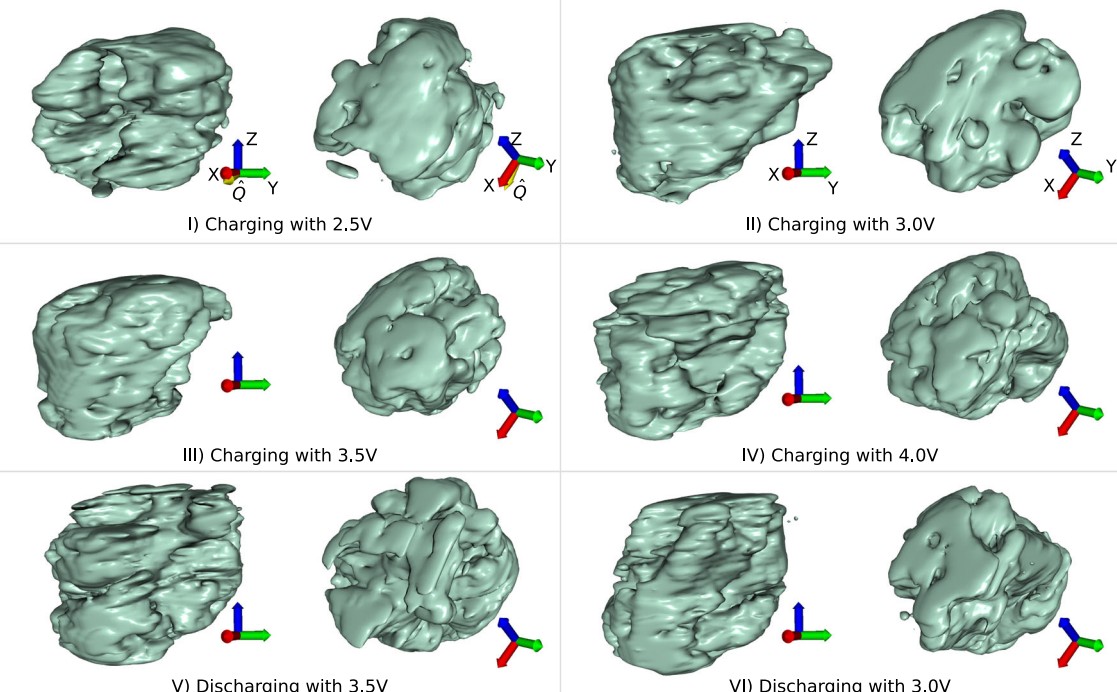

I) Charging with 2.5V

II) Charging with 3.0V

III) Charging with 3.5V

IV) Charging with 4.0V

V) Discharging with 3.5V

VI) Discharging with 3.0V

**Fig. 2 | Morphological reconstructions of the cycled LCO nanocrystal.** Reconstructions of the same crystal's diffractions from all the different charging and discharging stages visualised from two different directions to showcase the distinct reconstructions' morphologies. Generally, the reconstructions show a crystal of ~1.6 μm length, 1.4 μm width and 1.3 μm height visualised at an isosurface containing ~99.9% of the reciprocal data. The Cartesian axis in the bottom right corner of each object are identical throughout each vertical set, and assist with the orientation of the displayed objects from two different directions. Subfigure I (2.5 V charging state) shows the scattering vector's direction in the Cartesian axis ($\hat{Q} = 0.969\hat{x} - 0.249\hat{z}$).

along the scattering vector only (S6.iii in the Supplementary), with a measured resolution of 49.5 nm. Therefore, the same phase variation (and colour map in Fig. 3) between $-\pi$ and $\pi$ corresponds to strains between $-0.0037$ and $0.0037$. Table 1 shows values that are relative to the 2.5 V, except for the $\chi^2$ metric which only represents the fidelity of each reconstruction compared to their respective experimental data.

The PCA plot is shown in Fig. 3b, describing the alignment of each state's reconstruction with the determined most significant statistical eigenstate, also known as the principal component. The principal component alignment follows a path that is correlatable with the charging states and/or the lattice stability. The principal component starts at 0 due to the Fig. 3aI charging with 2.5 V being the chosen reference phase state, against which all the other states' phase differences are calculated.

Finally, Fig. 4 shows an interpretation of where the most significant phase changes occur. The entire crystal's morphology is translucent and the

interpreted most significant phase changes areas are opaque. All the opaque shapes contain at least 99% of the data and predominantly form on the surface of the object, shifting from the corner corresponding to the lower-left domain in Fig. 3a at lower voltages to the upper-left domain from Fig. 3a at higher voltages.

## Discussion and conclusions

The calculated $\chi^2$ values for the reconstructions (Table 1) suggest a high amount of accordance, therefore, the machine learning algorithm has reconstructed the phase successfully. Comparing charging states, the same region can show missing density in some regions while reconstructing well in others (Fig. 2). This is most likely due to the inhomogeneity of Li in that region that diffracts at different angles.

The domains that appear on the surface of the crystal throughout the charging reconstructions are almost exactly reproducible when discharging. The observed characteristic phase pattern can be due to a cumulation of screw and/or edge dislocations through and on the edges of the crystal[45]. The phase wrapping bordering the right-most domain has a specific position when the lattice is more stable, assumed during charging with 2.5 V (Fig. 3aI) and with 4 V (Fig. 3aIV), but tilts by ≈35° under cycling stress. Therefore, the right-most domain shifts and migrates with cycling.

The bottom-left domain does not appear to form initially (Fig. 3aI), but with increasing voltages, it begins expanding from the edge towards the centre (Fig. 3aII and III). This expanse continues up to ≈650 nm. The lack of discernible strain from the stable charged state (Fig. 3aIV) in this region suggests that the bottom-left domain has been mostly depleted of Li and formed a front that forbids Li to be removed from the central domain. When discharging, the dislocations and strains begin to reappear from the central domain towards the edge, suggesting that Li begins to fill as before. However, the Li refilling during the first discharge cycle is almost as complete as before charging, where the most significant differences are observed as phase changes in relatively small-sized regions throughout the lattice, particularly in the bottom-left region (Fig. 3aV and VI).

## Table 1 | Calculated metrics of LCO reconstructions

| Charging state | Charging | | | | Discharging | |
|---|---|---|---|---|---|---|
| Applied Voltage (V) | 2.5 | 3 | 3.5 | 4 | 3.5 | 3 |
| $\chi^2$ | 0.260 | 0.250 | 0.241 | 0.311 | 0.352 | 0.342 |
| $\Delta\phi_{mean}$ (deg) | 0.0 | 13.8 | 45.5 | $-1.1$ | 22.0 | 21.1 |
| Mean Relative Displacement (Å) | 0.0 | 0.142 | 0.467 | $-0.011$ | 0.226 | 0.217 |
| Mean Relative Strain (‰) | 0.0 | 0.57 | 1.87 | $-0.04$ | 0.90 | 0.87 |

Calculated $\chi^2$ values for each of the reconstructions against respective initial diffraction patterns. Mean phase difference relative to the reconstruction of the state when charging with 2.5 V, respective calculated average relative displacement compared to the state when charging with 2.5 V, and mean strain permillage relative to the state when charging with 2.5 V. All relative measurements are components along the scattering vector (calculations in S6.iii in the Supplementary).

The upper-left domain (Fig. 3aI) possibly existed from the beginning (phenomenon also observed by Estandarte et al.[22]) since it appears in the first charging stage (Fig. 3aI) and, with charging, it expands towards the central domain (Fig. 3aII and III), similarly to the lower left domain, but up to ≈200 nm in depth. At some point right before the charged state is achieved (Fig. 3aIV), the upper-left domain disintegrates into multiple smaller domains. It does not show signs of shape-reformation when discharging begins (Fig. 3aV); however, the observed strain begins to change compared to Fig. 3aIV's charged state. The last discharge shows the upper-left domain regaining its shape almost completely, with minor impurity-like phase changes throughout, suggesting that Li has not returned optimally. We associate this phenomenon to a plastic deformation from which the crystal cannot recover easily with the used cycling protocol.

The PCA graph (Fig. 3b) presents a certain degree of reproducibility of the discharging states versus the charging ones. As from the previous discussion, the plastic deformity that occurs between charging with 3.5 V and with 4 V is also possibly suggested from Fig. 3b. A noticeable increase in order parameter that we associate with instability also destabilises the configuration when discharging with 3.5 V. This is also confirmed by the domains failing to reform or migrate in Fig. 3aV compared to Fig. 3aIII. Also, the inverse PCA procedure (Fig. 4) showcases that the most amount of phase changes that are present in the reconstructions and that influence the actual PCA occur in the regions predominantly to the left side of the nanocrystal in Fig. 3a; i.e., upper-left dissipation and lower-left migrations generate the most amount of phase variations.

Viewing the charging and discharging with 3.0 V states from Figs. 3b and 4 in tandem suggests that the (approximate) 3.0 V state is a critical point for the crystal lattice strain along any charging or discharging paths. Therefore, the strain does not exactly correlate linearly with the applied voltage, but seems to follow the PCA graph of Fig. 3b instead.

All the edge domains have in common a migratory or expansive tendency in the incipient stages of cycling, but, it is possible that the 4 V charging stage has minimally altered the lattice in the edge domains. Such alterations can either improve the cyclability by building Li pathways to the inner regions, or, with numerous repeated cycles, induce damage to the lattice that renders it much less electrochemically active. The existence of a cycling protocol that abuses the crystals less, or promotes lattice self-healing cannot be excluded.

Our findings are similar to the observations from an experiment performed by Liu et al.[23], where they have analysed the domain dynamics inside Li- and Mn-rich cathode materials, such as $Li_2MnO_3$. They have observed that the crystal lattice expansion was confined by inactive material generating tensile strain that began to expand gradually with the voltage towards the inner nanocrystal. Furthermore, at a maximum voltage of 4.43 V, the nanocrystal's domains are all delithiated[23], much like in our experiment at the lower 4 V. Liu et al.[23] also note that the strain generated at the higher voltage could be due to the oxygen build-up, therefore, affecting the structural stability throughout the whole nanocrystal, possibly resulting in the domain collapsing.

Thus, the domain dynamics within LCO are not much different compared to the ones observed in $Li_2MnO_3$. Liu et al.[23] have unfortunately lost the crystal after charging it to 4.51 V, and results when discharging could not be obtained. Knowing that the replacement of Co with Mn results in cathode materials with higher capacities but less stability, it is expected that the lower voltages are enough to charge LCO faster to a smaller capacity, yet $Li_2MnO_3$ is less stable, which explains the crystallite's disappearance.

Our experiment's nanocrystal's size possibly hinders the Li being easily displaced from the central domain. On the other hand, the voltage application times (5 min) might be too short to allow Li displacement from the

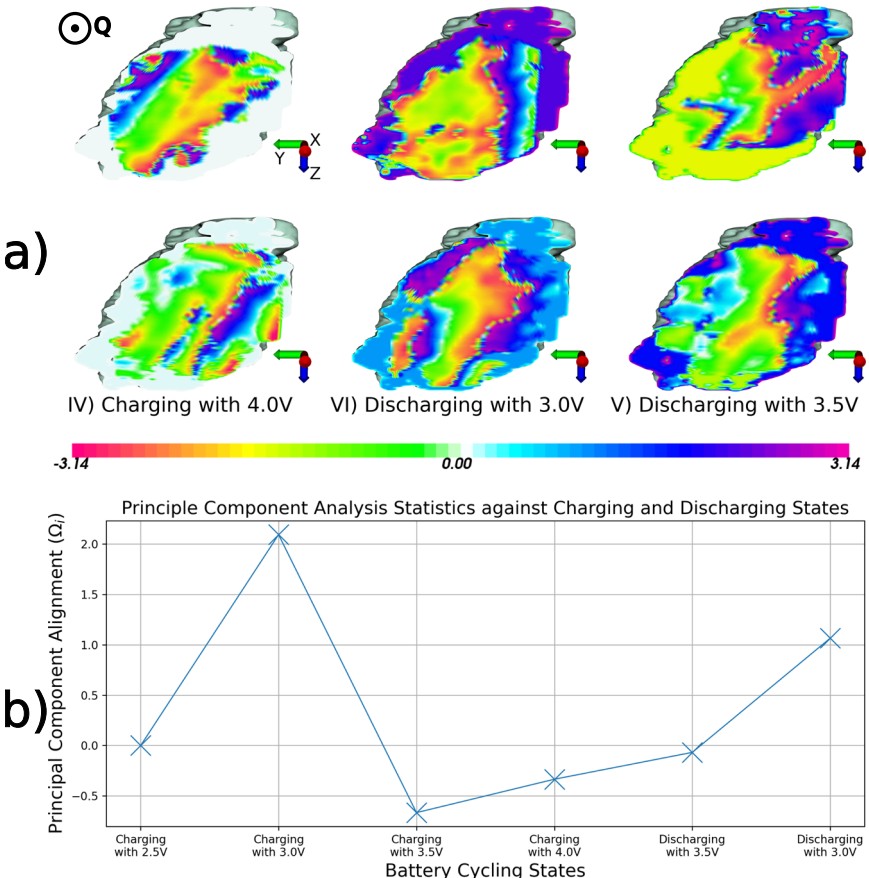

**Fig. 3 | Phase Information within the cycled LCO nanocrystal. a** Cross-sections of the reconstructions displaying the phase information within. Common phase variation features appear at certain positions within the lattice when charging and discharging. The maximum applied voltage produces phase patterns similar to the first charging stage, possibly showing some strain relaxation due to the lattice stabilising with fewer Li ions. The scattering vector's direction is approximately parallel to coming out of the page ($\hat{Q} = 0.969\hat{x} - 0.249\hat{z}$). The status proximity between Charging with 2.5 V and with 4.0 V is further confirmed by (**b**) the Principal Component Analysis results, where the stages with higher voltages are closer to the reference state.

**Fig. 4 | Most significant phase changes.** Visualisation of the most significant phase changes influencing the PCA. The appearing objects are at an isosurface containing between 99% and 99.7% of the data. The Cartesian axis in the bottom right corner are identical throughout each vertical set, and assist with the orientation of the displayed objects from three different directions.

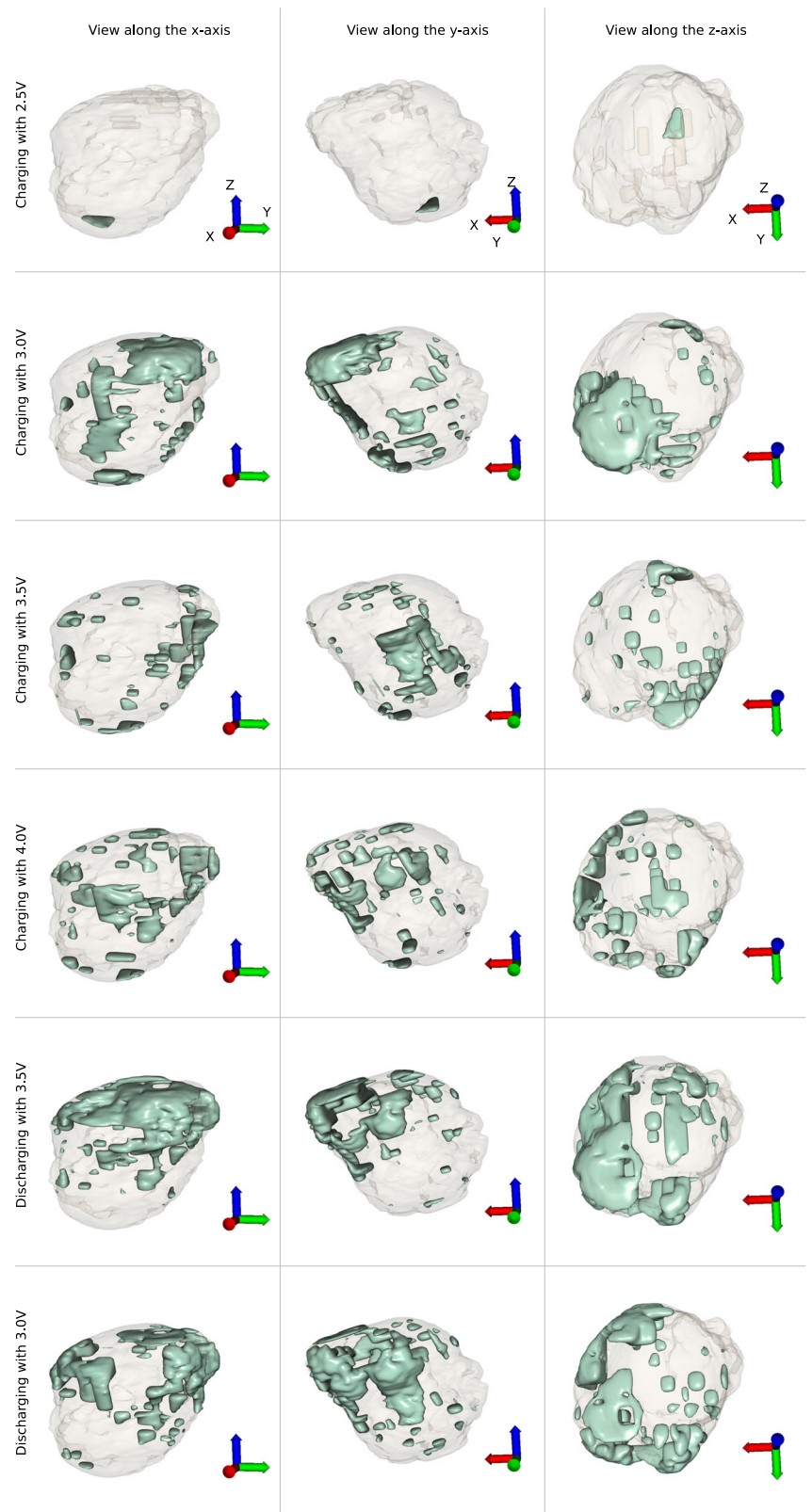

central domain, despite being enough for the measured current to drop significantly—confirming achieved charging/discharging status. The effects from longer times of voltage applications are unknown.

Therefore, the behaviour of domains during battery cycling can be classified into two: degenerative and migratory. When the voltage first starts to increase, the domains begin to form, extend, and migrate. The extent seems to reach a maximum of a few hundred nanometres, as seen in Fig. 3aIII and V. Therefore, theoretically, a nanocrystal of such dimensions should have most of its Li ions displaced to the exterior with minimal domain formation in the crystal's core and thus forbids the core Li entrapment. All the previously described domains from Fig. 3a show this behaviour. However, once a certain voltage is achieved, some domains can

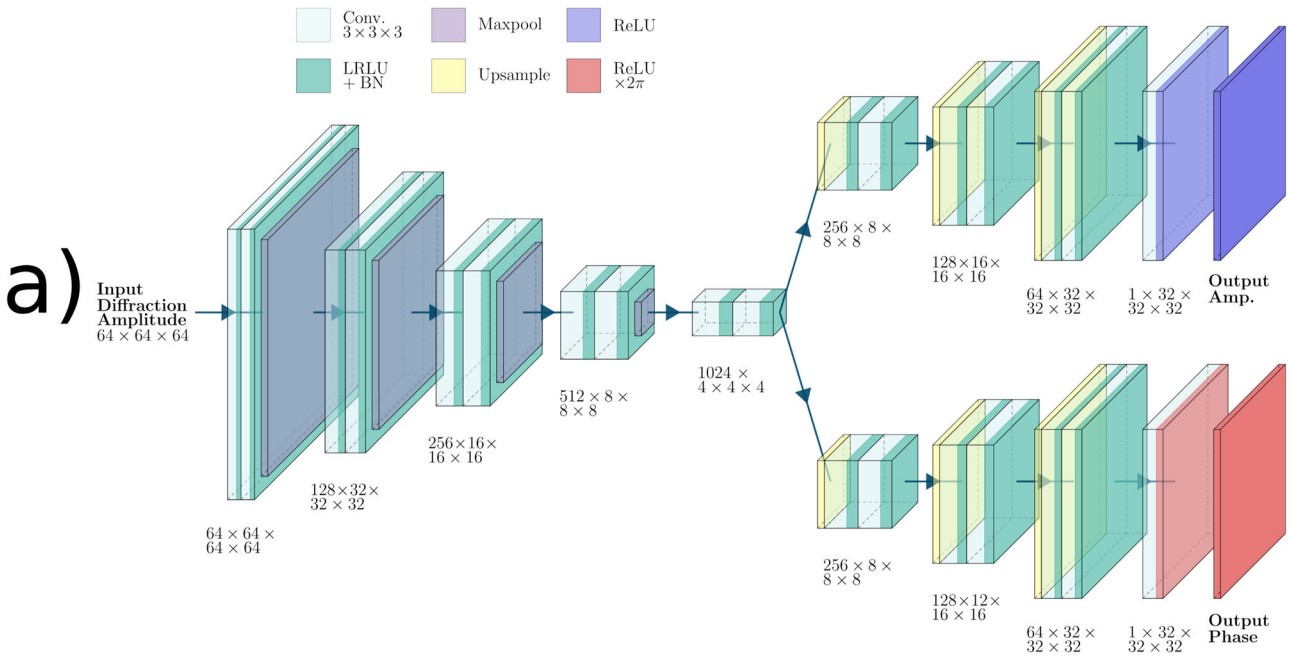

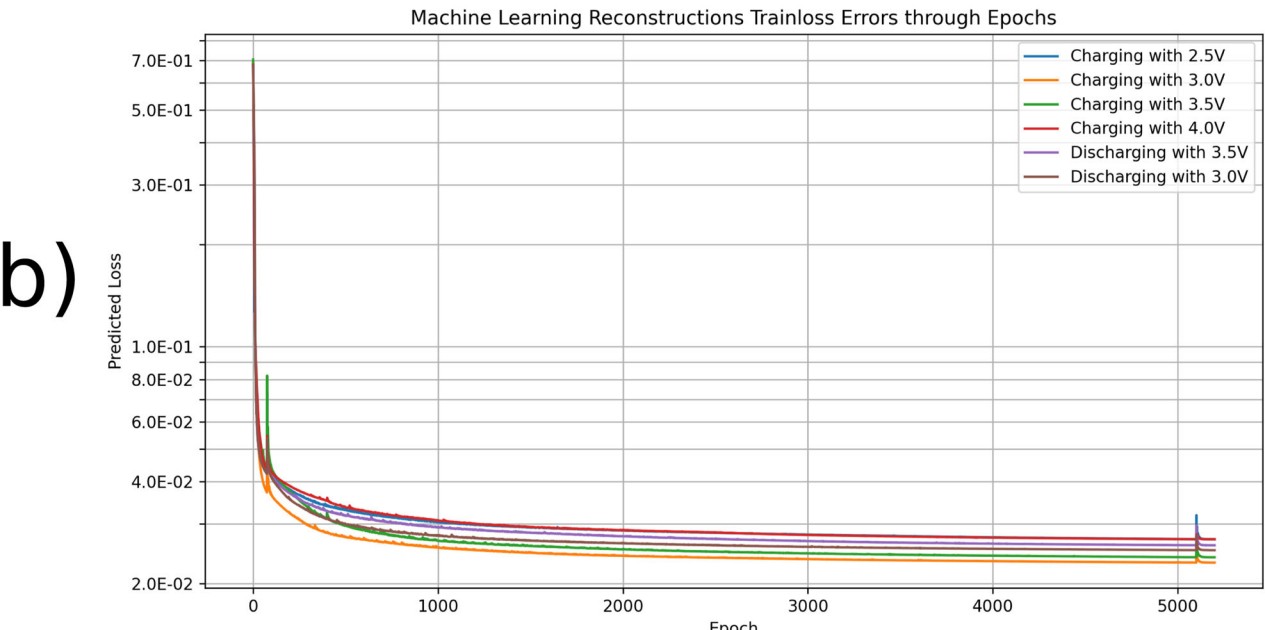

**Fig. 5 | Reconstructions' Machine Learning Algorithm. a** The convolutional neural network that uses an encoder-decoder framework to reconstruct the nanocrystals' morphologies and phase information. **b** Plots displaying the training loss data while using the neural network to reconstruct each of the simulated diffraction patterns of hexagonal systems against the number of epochs since the algorithm's beginning.

collapse by forming multiple dislocations within and multiple domains aligned in disorganised manners (Fig. 3aIV's upper left edge).

In conclusion, Bragg CDI was used to successfully image the *in-operando* domain formation, extension, degeneration and contraction within a LCO nanocrystal inside a coin cell during a first battery cycle, revealing the nano scale depth from which Li can be diffused when voltages are applied and how its return to the lattice can be affected by further domain degradation. As far as we know, a Bragg CDI reconstruction of a single LCO cathode nanocrystal as it undergoes charging and discharging structural changes is unprecedented. The reconstructions were obtained by using our effective machine learning phase retrieval algorithm. Slices of the reconstructions show the phase shifts within as the nanocrystal was charged and discharged. A PCA resulted in a set of components, one for each state,

describing the phase differences behaviour within the nanocrystal reconstructions with a single metric, clearly showing some degree of reproducibility of the charging states when discharging; but also concerns related to plastic deformations of domains that could alter the structure significantly in time such that the nanocrystal would be rendered obsolete. An inverse PCA has shown the regions that influence the PCA the most, confirming that the most of the phase changes occur on the surface, point noticed during the interpretation of the slices through the reconstructions.

Generally, the domains that form while charging extend up to a few hundred nanometres within the nanocrystal, where the delimiting dislocations tend to disappear as the charge state and a new stable lattice configuration are achieved. This work shows that some domains can disintegrate into multiple smaller domains delimited by shorter dislocations

that can form lasting pathways for the Li from the core of the nanocrystal to be displaced. When discharging, the domains shrink as some dislocations migrate towards the edges of the nanocrystal, while most of the domains resulting from disintegration condense belatedly and reform the previous domain to a high degree, except for the newly appeared long-lasting dislocations that can act as pathways for the core Li ions. The origin of the disintegration of domains is most likely the highest applied charging voltage. The further evolution of the domains structure within the nanocrystals during later cycles is unclear.

## Methods

### Coin cell design

The experiment was performed on a novel coin cell design where the X-rays are allowed to pass through a window to peer at cathode nanocrystals while they are being charged or discharged. The window of ~1 cm² acts as a substrate and consists of a layer of 300 μm of X-ray-transparent Kapton on which a layer of 2.2 μm Al is coated. LCO nanocrystals are spin-coated onto the Al surface which allows good electrical conductivity, while the much thicker Kaptop layer ensures good mechanical properties for mouting within a compressed cell (Fig. 1b). An Al tape frame ensures the connection between the Al layer facing the inner cell and the exterior steel shell. To ensure adequate LCO distributions on the Al layer, LCO nanocrystals were spin-coated. The experiment required to determine the predisposed alignment of the nanocrystal on the surface initially, but, during the Bragg CDI stage, lower densities are preferred as these would allow the beam to irradiate fewer nanocrystals at the same time. This was done with Pulse Laser Deposition using a shadow mask to obtain regions of higher and lower densities. Furthermore, only spin-coated LCO resulted in poor adhesion to the surface, risking the crystals rotating in the beam (in-detail preparation in S1 in Supplementary).

Finally, the substrate, which became essentially a low-capacity cathode, was submerged in the electrolyte along a Celgard 2400 monolayer microporous membrane acting as a separator, paired with a graphite anode, and clamped within a coin cell's caps that had a 5 mm diameter drilled hole in the centre. The hole is a frame through which the Kapton layer is visible and, therefore, a window for X-rays towards the LCO (Fig. 1b).

### Experimental setup

The Bragg CDI measurements were performed on the I13-1 beamline at Diamond Light Source synchrotron facility. Beamline I13-1 has an Xcalibur area detector situated at 2.8 m away from the sample, on a robotic arm with a maximum scattering angle of 30°. Additionally, the window of the coin cell restricted the minimum scattering angle to ≈15°. Furthermore, lower orders' reflections of LCO are in the proximity of the reflections from other coin cell components (such as Al), and the most isolated specular reflection was chosen as the $(1, 0, -5)$. Therefore, the beam's energy was selected by bringing the $(1, 0, -5)$ reflection within the allowed interval: a beam of an energy of 13.5 keV aimed directly at the coin cell window, returning towards the detector at $2\theta = 28.8°$. Therefore, in this experiment, LCO nanocrystallites were selected with a large scattering intensity and clear diffraction fringes at $(1, 0, -5)$.

The incident beam was reduced to 40 μm after isolating a suitable crystallite. The crystal was placed in the eucentric point and the detector collected frames of the nanocrystal's diffraction patterns in Bragg CDI rocking curve scans; rocking $\theta$ (pitch) in increments of 0.005°, covering a pitch range of 0.8° (Fig. 1a).

A Keithley 2410 Sourcemeter connected to the electrodes of the coin cell supplied an adjustable voltage and measured the charging/discharging currents. The current provided a measure of the cycling rate and status of the coin cell. A total of six voltages were applied to replicate the coin cell cycling in the following sequence: charging with 2.5 V, 3 V, 3.5 V and 4 V, and then discharging with 3.5 V, and 3 V. Each voltage was applied for 5 min (current measurements in S2 of the Supplementary material). The power source was then turned off and set into a high impedance mode while a total of 22 Bragg

CDI rocking curve scans collected the crystal's diffraction patterns at each cycle stage.

### Phase reconstructions

The phase retrieval was performed by using a deep learning model based on the Convolutional Neural Network (CNN) proposed by Wu et al.[46]. To obtain the reconstructions, we used an nVidia A100 GPU with 80 GB of VRAM. Each reconstruction took ~70 min, averaging to ~14 min for every 1000 epochs. The CNN used an encoder-decoder framework that initially encoded the measured amplitude in a feature space. The encoding was then split into two separate paths for later recovery: one for the amplitude, another for the phase[47]. The size of the output array was made to be half the size of the input diffraction pattern. Figure 5a illustrates this CNN. Training was performed using 30,000 Fourier pairs with hexagonal symmetry but random aspect ratios and a Gaussian-correlated phase profile, as described in refs. 46,[48]. This resulted in consistently low losses (less than $10^{-2}$) in up to 150 epochs when using the ADAM optimiser[49].

Using the trained network, we employed transfer learning where the experimental diffraction pattern is used to further train the pre-trained network across 5,000 epochs while simultaneously generating a prediction. The optimisation during this phase was guided by a loss metric that compares the Fourier transform of the predicted object with the provided diffraction pattern amplitude[46,48]. Figure 5b shows a plot of the resulting transfer learning loss.

Each set of 22 Bragg CDI rocking curves were summed together to form six diffraction patterns (one per charging state) and were fully prepared for the deep learning model, including identical real-space constraints (same crystal throughout the experiment). This resulted in a set of six objects corresponding to the different charging states. The variation in amplitude and phase of the CNN reconstructions was vanishingly small and below the resolution of our measurement after 5 iterations.

The $\chi^2$ metric described in Eq. (1) is one way to determine the fidelity with which the machine learning algorithm has reconstructed the nanocrystal at each charging stage ($\chi_s^2$ for state $s$). $\chi^2$ compares the values of the measured diffraction $|D_s^i|$ with the Fourier transform of the reconstructed $|R_s^i|$ data.

$$\chi_s^2 = \frac{\sum_i \left( \sqrt{|D_s^i|} - \sqrt{|R_s^i|} \right)^2}{\sum_i |R_s^i|}, \forall s \qquad (1)$$

A series of operations applied to each of the reconstructions has successfully resulted in identical morphologies (S3 in Supplementary), allowing for a comprehensive phase analysis. To determine the true morphology of the crystallite, a set of coordinate transformations returns the positions of the reconstructions' voxels in the laboratory frame of reference[50].

Subsequently, an insightful PCA was performed on the phase arrays of the reconstructions (S5 in the Supplementary)[51–53]. The PCA reduced the data dimensions while preserving the important information, resulting in a more easily interpretable analysis. The phase information was used to determine the experiment's Covariance Matrix ($C$) from Eq. (2), measuring the correspondence between the changes among variables: i.e., the phase differences' outer product for all charging states relative to the first charging state at 2.5 V ($\Delta\phi_i = \phi_i - \phi_0$, where $\phi_0$ is the phase when charging with 2.5 V).

$$C = \sum_i \Delta\vec{\phi_i} \otimes \Delta\vec{\phi_i} \qquad (2)$$

The above covariance matrix had N eigenvalue-eigenstate pairs ($C\vec{\psi_j} = \lambda_j \vec{\psi_j}$). The principal component ($\psi_1$) is the eigenstate belonging to the largest eigenvalue ($\lambda_1$)—directly related to the data variances. PCA determines the alignment of any of the experimental states to the principal component, i.e., scalar products between the states and the principal component ($\Omega_i$) and quantify the reconstructions' contributions to the total

structural variation (Eq. (3)). The order parameters resulting from the PCA were a direct representation of the phase variation changes within the nanocrystal compared to the initial state (charging with 2.5 V).

$$\Omega_i = \overrightarrow{\psi_1} \cdot \overrightarrow{\Delta\phi_i}, \forall i \tag{3}$$

However, another procedure allows the determination of the exact regions which influence the PCA the most ($\Phi_i$) in Eq. (4).

$$\overrightarrow{\Phi_i} = \overrightarrow{\Delta\phi_i} - \overrightarrow{\psi_1}, \forall i \tag{4}$$

Dividing the local phases by the magnitude of the **Q**-vector yields in the component of the atomic displacement along the scattering vector. The strain within the crystals can be estimated using Eq. (5) to approximate the strain along the Q vector.

$$\epsilon_{ij} = \frac{1}{2}\left(\frac{\partial \mathbf{u}_i}{\partial x_j} + \frac{\partial \mathbf{u}_j}{\partial x_j}\right) \tag{5}$$

where $\epsilon$ is the strain tensor.

## Data availability
The data underpinning the findings of this study are available from M.C.N. upon reasonable request.

## Code availability
The code for the phase retrieval algorithm is part of 'Bonsu: The Interactive Phase Retrieval Suite' software package, and can be downloaded through PyPI (https://pypi.org/project/Bonsu/) or GitHub (https://github.com/bonsudev/bonsu). The same Bonsu code was used and modified slightly to enable capturing the inverse PCA images with the opaque and translucent objects, and to automate the process of obtaining the images that formed the frames in the supplementary videos. The analysis itself consisted of some mathematical operations performed on the initial reconstruction and split in separate iterating python scripts and are available from M.C.N. upon reasonable request.

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

## Acknowledgements

This work was supported by United Kingdom Research and Innovation (UKRI) grant MR/T019638/1 for the University of Southampton Department of Physics & Astronomy and Diamond Light Source. The authors thank Diamond Light Source for the allocated time on Beamline I13-1 under the MG31929 proposal, as well as Beamline I16 for the NT34075, MM34025, MM29880 and MM27621. We acknowledge the use of the IRIDIS High-Performance Computing Facility and the associated support services at the University of Southampton. Finally, the authors thank Dr Chun Huang for her contribution during some of the earlier experiments at Diamond Light Source.

## Author contributions

M.C.N. designed and supervised the study. All authors contributed to the Bragg CDI experiment at Diamond Light Source. S.U. has supplied the cathode material. D.S. has prepared the cathode. M.Z. has prepared the anode and assembled the coin cell. D.S., M.C.N., D.G.P., S.P.C. and M.N. have personally assisted during the experiment at Diamond Light Source. A.B. has allowed the use of the Keithley sourcemeter, as well as other previous experiments at I16. P.L. and C.R. are part of beamline I13 and have given invaluable advice during the experiment. A.H.M. and M.C.N. developed the machine learning model for the crystal's reconstructions. D.S. performed data reconstruction and analysis. D.S. wrote the manuscript and supplementary material with input from all the authors.

## Competing interests

The authors declare no competing interests.
