## [Peer Review file · Communications Chemistry]

Imaging in-operando LiCoO₂ Nanocrystallites with Bragg Coherent X-ray Diffraction

Corresponding Author: Mr David-Alexandru Serban

Version 0:

Reviewer comments:

Reviewer #1

(Remarks to the Author)

The manuscript entitled "Imaging in-operando LiCoO₂ Nanocrystallites with Bragg Coherent X-ray Diffraction" explores the evolution of the crystal structure and nanoscale domains of LiCoO₂ during the initial lithium extraction and insertion process using the Bragg Coherent X-ray Diffraction method. This method demonstrates significant novelty and potential for investigating cathode materials. However, the logical analysis in this paper is weak, and evidence is lacking. Therefore, we do not recommend its acceptance at this time. We suggest that the author makes further substantial revisions. The detailed questions and suggestions are listed below.

1. The authors mentioned, "The missing densities within isolated reconstructions (Figure 2) could be due to defects not reflecting" on page 7. Will the existence of defects affect subsequent test and analysis? Please explain.
2. In the manuscript, "However, the Li refilling is not happening as completely as before due to small observable impurity-like phase changes throughout the lattice in the bottom-left region when discharging (Figure 3.a.V and VI)." on page 9. The author believes that the lithium intercalation is insufficient. What are the corresponding charge and discharge capacities? Please provide the actual specific capacity-voltage curve from your tests. If the situation is indeed as the author describes, the corresponding Coulombic efficiency should be very low. If this does not match, then the particles selected by the author are not representative, and the conclusions drawn are not universally applicable.
3. The authors mentioned, "All the edge domains have in common a migratory or expansive tendency in the incipient stages of cycling, but, it is possible that the 4 V charging stage has damaged the lattice in the edge domains and their likelihood to accept Li optimally." on page 9. A voltage of 4.0 V is not considered high for LiCoO₂, and it generally has good cycling stability. The authors suggests that the 4 V charged state has already damaged the crystal structure of the material, affecting the possibility of Li⁺ reinsertion. Does this lack sufficient statistical significance? Can a single particle truly represent the overall behavior of LiCoO₂?
4. The authors mentioned, "On the other hand, the voltage application times (5 minutes) might be too short to allow Li displacement from the central domain, despite being enough for the measured current to drop significantly - confirming achieved charging/discharging status." on page 10. Does the author believe there are issues with their experimental design? Please provide more experimental details, including specific capacity-voltage curves, voltage-time curves, and current-time curves. Additionally, how can this problem be further addressed to make the article more convincing?
5. In the manuscript, "All the previously described domains from Figure 3a show this behaviour. However, once a certain voltage is achieved, some domains can collapse by forming multiple dislocations within and multiple domains aligned in disorganized manners (Figure 3.a.IV's upper left edge)." on page 10. The author believes that some domains can collapse under such low voltage conditions. It would be helpful to use direct characterization methods, such as TEM, to further illustrate this issue.

Reviewer #2

(Remarks to the Author)

This is a nice result, the authors provide a detailed analysis of the evolution of their LiCoO₂ sample through multiple charge states by incorporating a deep neural network for the 3D BCDI reconstruction process. I believe that this work will be of interest to the community and the wider field.

I recommend that the paper be accepted for publication, below are some minor comments/questions for the authors to consider.

- 1). Figure 2 shows individual reconstructions for various charge/discharge states of the sample. I am curious how reproducible the reconstructions are for a given single charge state. Can you run your reconstruction process for I) - VI) in Figure 2 something like 100 times again and again for each of those cases and quantitatively summarize the variance of the predictions for each state? For example, maybe choose the first reconstruction for each state and then produce 99 more reconstructions and show your PCA graph where you are comparing multiple reconstructions of each given charge state to themselves, rather than comparing the reconstructions of different charge states to themselves. Maybe just show the variance of the principle component of each charge state for 100 different reconstructions of that one state.
- 2). In Figure 3b) why does the lower voltage charging at 3.0 V have the largest difference relative to the 2.5 V state?
- 3). Why is the Chi squared error of the 2.5 V state not zero in Table 1, aren't all of these calculations made relative to that state?
- 4). PCA is first used on page 7, but is not defined until page 14.
- 5). Relative to Figure 5, what GPUs were used for this re-training to reconstruct each charge state and how long did this take in real-world time?

Reviewer #3

(Remarks to the Author)

This paper utilizes the Bragg Coherent X-ray Diffraction (BCDI) method to study the LiCoO₂ (LCO) cathode material, which has been widely used in commercial lithium-ion batteries. The successfully reconstructed phase images indicate the formation of the strained domains. These findings imply the inefficiency-increasing domain dynamics within the batteries during cycling. The paper seems to provide useful information on the mechanical properties of the LCO batteries. It is well-written and organized. This reviewer only has a few minor comments.

1. The authors used LCO particles underneath Kapton to perform BCDI measurements. Since Kapton is not a conductor, it is possible that the LCO particles illuminated by the X-rays were not active. Though the 2th angle is provided in the SI, I suggest that the authors calculate the corresponding lattice information of the BCDI data to show that the LCO particles were active during cycling. The authors may also confirm this with in-situ XRD measurements if possible.
2. Fig. 2 and 3 present the phase (or strain) information of LCO particle using its (1, 0, -5) Bragg peak. However, the peak direction is missed in the Figure.
3. During cycling, the LCO particle tends to form strain domains as presented in Fig. 4 and Table 1. Which region of the particle leads to a wider strain distribution (i.e., from the surface, or from the inner part of the LCO particle)? The authors can use the histogram of the strain distribution to show this trend.
4. The paper discussed a lot of the strain information of the LCO particle, but all the BCDI results are about the phase information (i.e., displacement of the lattice). What does the 2D strain image look like?

Version 1:

Reviewer comments:

Reviewer #1

(Remarks to the Author)

The proposed problems have been well addressed, so the revised manuscript can be accepted.

Reviewer #2

(Remarks to the Author)

The authors have answered all of my questions and have added additional explanations to the paper. I am happy with the paper and recommend that it is accepted for publication.

Reviewer #3

(Remarks to the Author)

This Bragg CDI method demonstrates significant novelty for investigating cathode materials. The authors have addressed all my questions. Thus, I recommend publishing the paper.

Dear Editor and Reviewers,

We greatly appreciated your efforts and all your insightful comments and suggestions. Please find below a detailed response to all of your comments and suggestions. With additional data shown and many changes to the text and figures, we believe that these changes offer a substantial improvement to the manuscript and I hope that the responses provided below favourably answer your queries.

Our responses are drawn with blue ink, such as this letter. Texts from the manuscript are shown with an *italic* font in gray, removed text in red strike-through, and additional text in green. Both the manuscript and the supplementary material contain these edits with the same colouring rules.

We hope that you will now find the revised manuscript acceptable for publication in Nature Communications Chemistry.

Thank you very much.

Kindest regards,

David Serban (on behalf of all co-authors)

Reviewer 01

The manuscript entitled “Imaging in-operando LiCoO₂ Nanocrystallites with Bragg Coherent X-ray Diffraction” explores the evolution of the crystal structure and nanoscale domains of LiCoO₂ during the initial lithium extraction and insertion process using the Bragg Coherent X-ray Diffraction method. This method demonstrates significant novelty and potential for investigating cathode materials. However, the logical analysis in this paper is weak, and evidence is lacking. Therefore, we do not recommend its acceptance at this time. We suggest that the author makes further substantial revisions. The detailed questions and suggestions are listed below.

We thank the reviewer for their diligent work reviewing the paper and we are glad that they point out the significant novelty and potential for this technique. We have responded to each of the comments below and believe this has substantially improved the manuscript.

1. The authors mentioned, “The missing densities within isolated reconstructions (Figure 2) could be due to defects not reflecting” on page 7. Will the existence of defects affect subsequent test and analysis? Please explain.

Apologies for the confusion, this is not what we meant to state here. Our point is that regions of the sample are producing diffraction at angles not captured by the measurement. Indeed, Bragg CDI analysis is sensitive to lattice defects; however, in order to observe these defects, they must contribute to the diffraction at or near the Bragg angle. We have modified that sentence in the text to make this clearer:

“~~The missing densities within isolated reconstructions (Figure 2) could be due to defects not reflecting as well as the bulk of the nanocrystal in different states. Comparing charging states, the same region can show missing density in some regions while reconstructing well in others (Figure 2). This is most likely due to the inhomogeneity of Li in that region that diffracts at different angles.~~”

2. In the manuscript, “However, the Li refilling is not happening as completely as before due to small observable impurity-like phase changes throughout the lattice in the bottom-left region when discharging (Figure 3.a.V and VI).” on page 9. The author believes that the lithium intercalation is insufficient. What are the corresponding charge and discharge capacities? Please provide the actual specific capacity-voltage curve from your tests. If the situation is indeed as the author describes, the corresponding Coulombic efficiency should be very low. If this does not match, then the particles selected by the author are not representative, and the conclusions drawn are not universally applicable.

Thank you very much for your suggestion. Our reply to this is two-fold:

- It is generally infeasible to provide specific capacity-voltage curves for a single nanocrystal in-operando as we are unable to make electrical contact inside of the battery coin cell. However, we can show instead a curve of the c-axis length variation plotted against the charging states from this nanocrystal. This is now shown in Figure 6 of the Supplementary Material section S3. Variation of the c-axis length as measured by the variation in scattering angle of the Bragg peak (SI Fig 6) is consistent with the small capacity variation we see from current measurements (SI Fig 3). The c-axis expands/contracts when charging/discharging i.e. when Li is removed/added.

To clarify this, we have plotted the crystal lattice c-axis length throughout our experiment in Section S3 “Peaks Shift”.

- We have shown using the variation in scattering angle that the Coulombic efficiency of the nanoparticle is similar to that of the cell; thus, we believe this particle to be illustrative of the

whole cell. Of course particle size effects (the variation shown in the SEM image in SI Fig 10) will change the surface area and reactivity of individual particles, but we expect the internal dynamics to be consistent. We have added a discussion on these aspects in the SI Section S10.

We have rephrased the above-cited statement to better reflect our understanding:

“ However, the Li refilling ~~is not happening as completely as before~~ during the first discharge cycle is almost as complete as before charging, ~~due to small observable impurity like~~ where the most significant differences are observed as phase changes in relatively small-sized regions throughout the lattice ~~in the bottom-left region when discharging~~ , particularly in the bottom-left region (Figure 3.a.V and VI).”

3. The authors mentioned, “All the edge domains have in common a migratory or expansive tendency in the incipient stages of cycling, but, it is possible that the 4 V charging stage has damaged the lattice in the edge domains and their likelihood to accept Li optimally.” on page 9. A voltage of 4.0 V is not considered high for LiCoO₂, and it generally has good cycling stability. The authors suggests that the 4 V charged state has already damaged the crystal structure of the material, affecting the possibility of Li⁺ reinsertion. Does this lack sufficient statistical significance? Can a single particle truly represent the overall behaviour of LiCoO₂?

Thank you very much for this comment. Apologies, our wording might have been misleading. We agree that 4 V is too low to create any (lasting or grave) damages to the lattice at this stage and have rephrased where necessary. We have only noticed minor alterations within the lattice, as discussed in the manuscript. These alterations might be favourable for the nanocrystal to be cycleable with Li from the inner core of the nanocrystal, by creating migration pathways from the inner regions. It is possible however that these minor alterations are the microscopic seeds for battery degradation over many repeated cycles, as we might expect these minor strain variations to evolve into cracks or defects, reducing performance.

“ [...], it is possible that the 4 V charging stage has ~~damaged~~ minimally altered the lattice in the edge domains ~~and their likelihood to accept Li optimally~~. Such alterations can either improve the cyclability by building Li pathways to the inner regions, or, with numerous repeated cycles, induce damage to the lattice that renders it much less electrochemically active.”

4. The authors mentioned, “On the other hand, the voltage application times (5 minutes) might be too short to allow Li displacement from the central domain, despite being enough for the measured current to drop significantly - confirming achieved charging/discharging status.” on page 10. Does the author believe there are issues with their experimental design? Please provide more experimental details, including specific capacity-voltage curves, voltage-time curves, and current-time curves. Additionally, how can this problem be further addressed to make the article more convincing?

We greatly appreciate your question, thank you. To be candid, imaging an inactive crystallite was not anticipated as it is an unlikely occurrence. However, this is related to the conclusion of our paper in itself. Therefore, after this experience, we have considered ways to improve the methodology and ensure success in future experiments. The time constraints of a beamtime at a national synchrotron are such that we were not able to afford much time to charging, however as shown by our reference cell measurements, now included in SI Fig 10-13, a charge time of 5 minutes is sufficient to charge the cell.

Regarding the plots you recommend we share, as noted above in point 2, it is generally infeasible to provide specific capacity-voltage curves for a single nanocrystal in-operando as we are unable to make electrical contact inside of the battery coin cell. We can however show characterisation measurements from a reference cell, made in exactly the same manner, but missing the open window. Current-time curves and Capacity-voltage curves are given in SI Figures 11-13. In addition, and as

described above, we have added a plot of the c -axis changing with charging state, illustrating the behaviour of the particle (SI Figure 6).

5. In the manuscript, “All the previously described domains from Figure 3a show this behaviour. However, once a certain voltage is achieved, some domains can collapse by forming multiple dislocations within and multiple domains aligned in disorganized manners (Figure 3.a.IV’s upper left edge).” on page 10. The author believes that some domains can collapse under such low voltage conditions. It would be helpful to use direct characterization methods, such as TEM, to further illustrate this issue.

Performing another direct characterisation method on the same nanocrystal is highly improbable because the coin cell contains countless such nanocrystals of which we only investigated one. As for the low voltage, yes, we agree 4 V is definitely low. However, we were investigating the crystal structure on the nanoscale. Therefore, whatever strains and domain dynamics that appear in the crystal can be substantial. This is just one nanocrystal of many and its observed effects might be dramatic for one, yet less apparent on larger scales. The results and discussions of Estandarte et al. [1] and Liu et al.[2]’s papers covering NMC811 and Li_2MnO_3 , respectively, (referenced in our manuscript) are supportive of our conclusions, as noted in the manuscript. We have included in the supplementary material an SEM image of a region from the cathode substrate, taken before the coin cell was assembled (SI Fig 10).

Reviewer 02

This is a nice result, the authors provide a detailed analysis of the evolution of their LiCoO₂ sample through multiple charge states by incorporating a deep neural network for the 3D BCDI reconstruction process. I believe that this work will be of interest to the community and the wider field.

I recommend that the paper be accepted for publication, below are some minor comments/questions for the authors to consider.

We thank the reviewer for their kind words. We have answered to each of their comments below and made changes and additions to our material in accordance.

1. Figure 2 shows individual reconstructions for various charge/discharge states of the sample. I am curious how reproducible the reconstructions are for a given single charge state. Can you run your reconstruction process for I) - VI) in Figure 2 something like 100 times again and again for each of those cases and quantitatively summarize the variance of the predictions for each state? For example, maybe choose the first reconstruction for each state and then produce 99 more reconstructions and show your PCA graph where you are comparing multiple reconstructions of each given charge state to themselves, rather than comparing the reconstructions of different charge states to themselves. Maybe just show the variance of the principle component of each charge state for 100 different reconstructions of that one state.

Excellent point! The reliability of the convolutional neural network (CNN) concerned us as well in the beginning. We limited the number of iterations to reconstruct each state only 5 times and the results were identical throughout each. Literally, $\sigma_{x,y,z}^s=0, \forall$ voxel coordinates x, y, z and \forall state s , where $\sigma_{x,y,z}$ is the standard deviation of the set of values of voxel (x, y, z) across all the iterations' reconstructions. Therefore, there is no randomness appearing through the reconstructions that could give any different results. We have added words to indicate this within the manuscript.

“ The variation in amplitude and phase of the CNN reconstructions was vanishingly small and below the resolution of our measurement after 5 iterations.”

2. In Figure 3b) why does the lower voltage charging at 3.0 V have the largest difference relative to the 2.5 V state?

The PCA values show how much the crystal is strained within. Figure 3a shows these strains in the 3.0 V reconstructions (both charging and discharging) to be stronger than in any other of the reconstructions. Because Figure 4 shows the regions of these most significant strains to be of similar size (particularly comparing the 3.0 V charged state with the 3.5 V discharged state), we deduce that it is the amount of strain that influences the component alignment the most. We have added the following to the main manuscript to make it clearer:

“ Viewing the charging and discharging with 3.0 V states from Figures 3.b and 4 in tandem suggests that the (approximate) 3.0 V state is a critical point for the crystal lattice strain along any charging or discharging paths. Therefore, the strain does not exactly correlate linearly with the applied voltage, but seems to follow the PCA graph of Figure 3.b instead.”

3. Why is the Chi squared error of the 2.5 V state not zero in Table 1, aren't all of these calculations made relative to that state?

The χ^2 error of a state is obtained by comparing the Fourier Transform of the reconstruction with their respective experimental diffraction pattern, without accessing any of the other states' data. We perform a Fourier Transform operation on the reconstructions to obtain the simulated diffraction pattern that would correspond to the reconstruction. Then, we take the absolute values at each

of the voxels and compare them to the voxels at the same positions in the experimental data. Therefore, χ^2 is independent of other states, including 2.5 V state as the values are not determined relative to this state. To make this clearer to the reader, we have added the following to the main manuscript:

“ Table 1 shows values that are relative to the 2.5 V, except for the χ^2 metric which only represents the fidelity of each reconstruction compared to their respective experimental data.”

“ Table 1: [...] respective initial diffraction patterns; ~~mean~~Mean phase difference relative to the reconstruction of the state when charging with 2.5 V, respective calculated average relative displacement compared to the state when charging with 2.5 V, and mean strain permillage relative to the state when charging with 2.5 V . All relative measurements are components along the scattering vector (calculations in S6.iii in the supplementary).”

4. PCA is first used on page 7, but is not defined until page 14.

Apologies. We added an introduction to the PCA in the first paragraph of the "Results" section, referencing the corresponding detailed description in the "Methods" section.

“ The principal component analysis (PCA) performed on the reconstructions to quantify the phase changes was performed as detailed in the Methods section.”

5. Relative to Figure 5, what GPUs were used for this re-training to reconstruct each charge state and how long did this take in real-world time?

Training typically takes tens of hours to multiple days. However, obtaining a result from a singular diffraction pattern takes approximately an hour and 10 minutes (or about 14 minutes per 1000 epochs). We have added this useful information to the Methods section:

“ To obtain the reconstructions, we used an nVidia A100 GPU with 80 GB of VRAM. Each reconstruction took approximately 70 minutes, averaging to approximately 14 minutes for every 1000 epochs. ”

Reviewer 03

This paper utilizes the Bragg Coherent X-ray Diffraction (BCDI) method to study the LiCoO₂ (LCO) cathode material, which has been widely used in commercial lithium-ion batteries. The successfully reconstructed phase images indicate the formation of the strained domains. These findings imply the inefficiency-increasing domain dynamics within the batteries during cycling. The paper seems to provide useful information on the mechanical properties of the LCO batteries. It is well-written and organized. This reviewer only has a few minor comments.

Thank you very much! Below you can find our replies to your comments and the modifications we brought to the material to inform the reader further.

1. The authors used LCO particles underneath Kapton to perform BCDI measurements. Since Kapton is not a conductor, it is possible that the LCO particles illuminated by the X-rays were not active. Though the 2 θ angle is provided in the SI, I suggest that the authors calculate the corresponding lattice information of the BCDI data to show that the LCO particles were active during cycling. The authors may also confirm this with in-situ XRD measurements if possible.

Thank you for your suggestion. We have rephrased the coin cell preparation section in an attempt to make it clear that the LCO crystallites are in contact with the evaporated Al surface on the Kapton, which is in contact with the Al frame that goes over the Kapton layer edge to the other side, which in turn is in contact with the steel cap of the coin cell. We have also calculated the lattice parameters from the 2θ angles of the peaks vs the states and added the results in the supplementary material. The following are the modifications to the manuscript.

“ The experiment was performed on a novel coin cell design where the X-rays are allowed to pass through a window to peer at cathode nanocrystals while they are being charged or discharged. The window of approximately 1 cm² acts as a substrate and consists of a layer of 300 μm of X-ray-transparent Kapton on which a layer of 2.2 μm Al is coated. LCO nanocrystals are spin-coated onto the Al surface which allows good electrical conductivity, while the much thicker Kapton layer ensures good mechanical properties for mounting within a compressed cell (Figure 1.b). An Al tape frame ensures the connection between the Al layer facing the inner cell and the exterior steel shell. LCO nanocrystals are spin-coated onto a 300- μm Kapton window with a 2.2- μm Al coating to ensure good electrical conductivity. Kapton is highly transparent to X-rays and provides good mechanical properties for mounting inside a compressed cell, as illustrated in Figure 1.b.”

2. Fig. 2 and 3 present the phase (or strain) information of LCO particle using its (1, 0, -5) Bragg peak. However, the peak direction is missed in the Figure.

We have added the approximate Q-vector direction (out of the page) in one of the cross-sections from Figure 3 and to the coordinate axis in the 2.5 V state from Figure 2 ($\hat{\mathbf{Q}} = 0.969\hat{\mathbf{x}} - 0.249\hat{\mathbf{z}}$). Also, we have specified the exact alignment of the Q-vector in terms of the three Cartesian axes in the images in the captions of both Figures 2 and 3.

“ Fig. 2:[...] Subfigure I (2.5 V charging state) shows the scattering vector’s direction in the Cartesian axis ($\hat{\mathbf{Q}} = 0.969\hat{\mathbf{x}} - 0.249\hat{\mathbf{z}}$).”

“ Fig. 3:[...] The scattering vector’s direction is approximately parallel to coming out of the page ($\hat{\mathbf{Q}} = 0.969\hat{\mathbf{x}} - 0.249\hat{\mathbf{z}}$). ~~The scattering vector (\mathbf{Q}) is approximately parallel to the X axis (red) from the Cartesian axes found at the bottom right corners (approximately out of the page’s plane).~~”

3. During cycling, the LCO particle tends to form strain domains as presented in Fig. 4 and Table 1. Which region of the particle leads to a wider strain distribution (i.e., from the surface, or from the

inner part of the LCO particle)? The authors can use the histogram of the strain distribution to show this trend.

We are sorry for the confusion and we thank you for the opportunity to clarify it. Figure 4 shows a visualisation of the most significant phase changes influencing the PCA, i.e. the regions with the greatest change in phase (or lattice parameters) when the battery is cycled. It suggests that the phase information that changes the most seems to be on the edges of the particle. This is also previously shown by Figure 3a which shows slices through the reconstructed objects, where the outer edges change the most. Figure 4 does not show phase information or domain information, but, instead, is derived from the PCA. Regarding the strain in the particle’s core, though there are some changes in phase to the inner domain’s edges as it interacts with the other domains, the strain within remains largely constant. We have comments in the manuscript to this effect in the discussion and modified to clarify:

“ Also, the inverse PCA procedure (Figure 4) showcases that the most amount of phase changes that ~~occur~~ are present in the reconstructions and that influence the actual PCA ~~are happening~~ occur in the regions predominantly to the left side of the nanocrystal in Figure 3.a; i.e., upper-left ~~disintegration~~ ~~dissipation~~ and lower-left migrations generate the most amount of phase variations.”

4. The paper discussed a lot of the strain information of the LCO particle, but all the BCDI results are about the phase information (i.e., displacement of the lattice). What does the 2D strain image look like?

We have determined the phase information, from which we directly calculated the displacements along the Q-vector. We cannot state more at this point about the true displacement, because we only have one component of the displacement field in the direction of the Q-vector. The strain we calculated was linearly determined from this singular component of the true displacement, therefore, not the full tensor field of the strain. Strain information along the direction of the Q-vector is shown in Table 1. We have emphasised this point in the manuscript to make it clearer:

“ Table 1 shows values that are relative to the 2.5 V, except for the χ^2 metric which only represents the fidelity of each reconstruction compared to their respective experimental data.”

“ Table 1: [...] respective initial diffraction patterns; ~~mean~~ Mean phase difference relative to the reconstruction of the state when charging with 2.5 V, respective calculated average relative displacement compared to the state when charging with 2.5 V, and mean strain permillage relative to the state when charging with 2.5 V . All relative measurements are components along the scattering vector (calculations in S6.iii in the supplementary).”

References

- [1] A.K.C. Estandarte et al. “Operando Bragg Coherent Diffraction Imaging of LiNi_{0.8}Mn_{0.1}Co_{0.1}O₂ Primary Particles within Commercially Printed NMC811 Electrode Sheets”. In: *ACS Nano* 15.1 (2021), pp. 1321–1330. DOI: 10.1021/acsnano.0c08575.
- [2] T. Liu et al. “Origin of structural degradation in Li-rich layered oxide cathode”. In: *Nature* 606 (June 2022), pp. 305–312. DOI: 10.1038/s41586-022-04689-y.